# Comparative Rumen Metagenome and CAZyme Profiles in Cattle and Buffaloes: Implications for Methane Yield and Rumen Fermentation on a Common Diet

**DOI:** 10.3390/microorganisms12010047

**Published:** 2023-12-27

**Authors:** Pradeep K. Malik, Shraddha Trivedi, Atul P. Kolte, Archit Mohapatra, Siddharth Biswas, Ashwin V. K. Bhattar, Raghavendra Bhatta, Habibar Rahman

**Affiliations:** 1ICAR-National Institute of Animal Nutrition and Physiology, Bangalore 560030, India; pradeep.malik@icar.gov.in (P.K.M.);; 2International Livestock Research Institute, South Asia Regional Office, New Delhi 110012, India

**Keywords:** buffaloes, cattle, diet, metagenome, methane yield, rumen

## Abstract

A study was undertaken to compare the rumen microbial community composition, methane yield, rumen fermentation, and CAZyme profiles between cattle and buffaloes. The primary aim of this study was to ascertain the impact of the host species on the above when diet and environmental factors are fixed. A total of 43 phyla, 200 orders, 458 families, and 1722 microbial genera were identified in the study. *Bacteroidetes* was the most prominent bacterial phylum and constituted >1/3rd of the ruminal microbiota; however, their abundances were comparable between cattle and buffaloes. Firmicutes were the second most abundant bacteria, found to be negatively correlated with the Bacteroidetes. The abundances of Firmicutes as well as the F/B ratio were not different between the two host species. In this study, archaea affiliated with the nine phyla were identified, with Euryarchaeota being the most prominent. Like bacterial phyla, the abundances of Euryarchaeota methanogens were also similar between the cattle and buffaloes. At the order level, *Methanobacteriales* dominated the archaea. Methanogens from the *Methanosarcinales*, *Methanococcales*, *Methanomicrobiales*, and *Methanomassiliicoccales* groups were also identified, but at a lower frequency. *Methanobrevibacter* was the most prevalent genus of methanogens, accounting for approximately three percent of the rumen metagenome. However, their distribution was not different between the two host species. CAZymes affiliated with five classes, namely CBM, CE, GH, GT, and PL, were identified in the metagenome, where the GH class was the most abundant and constituted ~70% of the total CAZymes. The protozoal numbers, including *Entodiniomorphs* and *Holotrichs*, were also comparable between the cattle and buffaloes. Results from the study did not reveal any significant difference in feed intake, nutrient digestibility, and rumen fermentation between cattle and buffaloes fed on the same diet. As methane yield due to the similar diet composition, feed ingredients, rumen fermentation, and microbiota composition did not vary, these results indicate that the microbiota community structure and methane emissions are under the direct influence of the diet and environment, and the host species may play only a minor role until the productivity does not vary. More studies are warranted to investigate the effect of different diets and environments on microbiota composition and methane yield. Further, the impact of variable productivity on both the cattle and buffaloes when the diet and environmental factors are fixed needs to be ascertained.

## 1. Introduction

After carbon dioxide (CO_2_), methane (CH_4_) is the second most abundant greenhouse gas [1] and is present in the atmosphere, with a concentration of 1911 ppb in 2022 [2]. Livestock contribute about 15% of anthropogenic greenhouse gas emissions [3], and enteric fermentation remains one of the largest sources of CH_4_ emissions in agriculture [4]. Cattle and buffaloes are two major enteric CH_4_ emitters and aggregately contribute more than 90% of the total enteric CH_4_ [5]. In India, about 7.83 Tg of CH_4_ is emitted annually from cattle and buffaloes only [6]. Apart from being responsible for global warming, CH_4_ emission due to enteric fermentation is also accountable for the loss of feed energy [7,8].

The reports on the ability of cattle and buffaloes in terms of fibre degradability and methane emission are not in agreement. Cattle have a greater abundance of *Methanobrevibacter* than buffaloes [9]. On the contrary, some of the recent studies comparing the methane yield and microbiota composition using 16S rRNA [10] or shotgun metagenome sequencing [11,12] did not reveal any measurable difference in archaeal communities between cattle and buffaloes. Previous reports suggest that buffaloes are better fibre digesters and possess a different microbial profile than cattle [9,13,14]. Contrastingly, a recent study by [15] reported a similar cellulolytic bacterial community between swamp buffalo and cattle when fed on a similar diet. The performance of the ruminant is dependent on the rumen’s ability to efficiently break down the structural carbohydrates in the diet (Barrett et al., 2022 [16]) and is accomplished by a specific group of microbial enzymes called carbohydrate-active enzymes (CAZymes) (Lim et al., 2013 [17]).

The contradictory reports on the superiority of buffaloes in terms of methane emission and rumen microbiota profile warrant more data generation through systematic studies where diet and environmental factors are kept constant to avoid any confounding effect on the host species. Therefore, the present study was undertaken to compare the rumen microbial community composition, methane yield, rumen fermentation, and CAZymes between cattle and buffaloes while fed on similar dietary regimes and feed ingredients, and maintained under uniform environmental conditions so that a conclusion can be drawn about whether the host really has an impact on the shape of the rumen microbial community and CH_4_ yield.

## 2. Materials and Methods

### 2.1. Ethical Approval

The study was conducted at the Experimental Livestock Unit of the ICAR-National Institute of Animal Nutrition and Physiology, Bangalore, India. Animal handling and sample collection procedures for enteric emissions and ruminal fluid were endorsed by the Institute Animal Ethics Committee (IAEC) constituted under the Committee for Control and Supervision of Experiments on Animals (CPCSEA), Ministry of Fisheries, Animal Husbandry, and Dairying, Government of India (approval no. NIANP/IAEC/1/2020/5).

### 2.2. Animal Feeding and Management

The study included adult male crossbred (*B. taurus* × *B. indicus*) cattle (BW 533 ± 24.8 kg, 6–7 years) and buffaloes (BW 308 ± 21.2 kg, 6–7 years), six each, to explore the differences in rumen metagenome, CAZymes, CH_4_ yield, and ruminal fermentation between two host species when fed on a similar diet. The confounding effect of diet, environmental factors, and managemental practices on the rumen metagenome was controlled by housing the animals in the same location. The housing shed was east–west oriented, with 1.8-m-high walls on the north and south sides and wire fences on the top. The shed was provisioned with individual feeding and watering facilities. All the animals were dewormed 10 days prior to the start of the experiment using fenbendazole at 5 mg/kg BW. All the experimental animals were fed *ad libitum* at 10:00 h and had access to clean drinking water throughout the day. The experimental diet consisted of finger millet (*Eleusine coracana*) straw, para grass (*Brachiaria mutica)*, and a concentrate mixture in equal proportions. The concentrate mixture was formulated using the following ingredients: maize grain (320 g/kg), wheat bran (400 g/kg), soybean meal (130 g/kg), groundnut cake (120 g/kg), mineral mixture (20 g/kg), and salt (10 g/kg). The entire feeding experiment was conducted over 45 days, while the samples for enteric CH_4_ measurement and digestibility trials were collected in the last 15 days.

### 2.3. Ruminal Digesta Collection

On the last day of the experiment, approximately 45 mL of ruminal digesta containing both the liquid and solid fractions was collected from the individual animals at 3 h post-feeding. The ruminal fluid samples were collected through a nylon stomach tube of 2 m length connected to a vacuum pump (Mityvac 8000, Lincoln Industrial Corp., St. Louis, MO, USA) via an airtight collection vessel at one end [18,19,20]. To avoid saliva contamination, the initial 30 mL of ruminal fluid was discarded, and the subsequent 45 mL of the fluid were collected for the study. The collected rumen digesta were divided into three subsets of 15 mL each for the isolation of metagenomic DNA (set 1), ammonia and volatile fatty acids (set 2), and protozoal enumeration (set 3). For metagenomic DNA isolation, the rumen digesta samples without filtration were individually transferred to 50-mL conical tubes (Tarsons) and placed in the ice box while transporting the samples to the laboratory. Set 2 of ruminal digesta samples was filtered through double layers of muslin cloth, and the filtrate was collected in another 50 mL conical tube and transported to the laboratory for the estimation of ammonia and volatile fatty acids (VFA). The rumen digesta samples to be used for the protozoal enumeration were transported to the laboratory without being placed on ice. To allow the settling of feed particles and microbiota, set 2 was centrifuged at 13,000 rpm at 4 °C for 15 min, and the supernatant was preserved for ammonia and VFA estimation. For the ammonia nitrogen, 2–3 drops of saturated HgCl_2_ were mixed with the other half of the supernatant obtained, while the metaphosphoric acid (25%) in a 4:1 ratio (*v*/*v*) was mixed with another supernatant to estimate VFA. The processed samples for the ammonia nitrogen and VFA estimation were preserved at −80 °C until analysis.

### 2.4. DNA Extraction

Following the RBB+C method of [21], metagenomic DNA was extracted from the ruminal digesta samples collected from cattle and buffaloes. Specifically, 0.25 g of ruminal digesta containing liquid and solid fractions was combined with 1 mL of lysis buffer in a 2-mL sterile screw-cap tube (BioSpec Products, Inc., Bartlesville, OK, USA). The sterile screw-cap tube contained 0.5 g of pre-sterilised zirconia beads of 0.1 mm diameter (BioSpec Products, Inc., Bartlesville, OK, USA). The samples in sterile screw-cap tubes were homogenised for 3 min at maximum speed with a mini bead beater (Biospec Products, Inc., Bartlesville, OK, USA) and then incubated at 70 °C for 15 min with intermittent mixing. After 13,000× *g* centrifugation, the supernatant was collected in a 2 mL Eppendorf tube. After adding 300 μL of lysis buffer to the remaining content in the screw-cap tube and repeating the bead beating procedure, the supernatant was added to the Eppendorf tube containing the previously collected supernatant. To precipitate proteins and polysaccharides, the supernatant was treated with 260 μL of 10M ammonium acetate and stored on ice for 5 min. In order to separate the supernatant, the content was centrifuged at 4 °C for 10 min at 13,000× *g*, after which an equal volume of isopropanol was added and mixed by inverting the tube, followed by 30 min of incubation on ice. The DNA was collected by centrifugation at 4 °C for 10 min at 13,000× *g*, after which the pellet was washed with 70% ethanol. The DNA pellet was redissolved in 100 μL Tris-EDTA buffer, and 2 μL DNase-free RNase (10 mg/mL) was added to remove RNA contamination, followed by 15 min of storage at room temperature. After that, 200 μL of AL buffer and 15 μL of proteinase K were added to the DNA preparation, and it was left at 70 °C for 15 min to get rid of protein contaminants. The DNA in the suspension was mixed with 200 μL of absolute ethanol and purified using the QIAamp DNA mini kit (Qiagen, Hildenberg, Germany) according to the manufacturer’s instructions. With the help of 0.8% agarose gel electrophoresis and an Invitrogen Qubit 4.0 fluorometer (Life Technologies, Eugene, OR, USA), the quality and concentration of metagenomic DNA were assessed.

### 2.5. Shotgun Metagenome Sequencing

The metagenomic DNA sequencing was performed on the HiSeq2500 (Illumina Inc., San Diego, CA, USA) platform at Eurofins Genomics in Bangalore, India. For metagenomic library preparation, the NEBNext^®^ UltraTM II FS DNA Library Prep Kit for Illumina^®^ (New England Biolabs, Ipswich, MA, USA) was employed. About 100–500 ng of DNA was fragmented to 350 bp using NEBNext Ultra II FS Reaction Buffer and Ultra II FS Enzyme Mix in a PCR thermal cycler, with the subsequent incubation steps: 30 min at 37 °C, 30 min at 65 °C, and hold at 4 °C. The fragmented DNA was ligated with the NEBNext Adaptor for Illumina by combining 35 μL of fragmented DNA with the NEBNext Ultra II Ligation Master Mix and incubating it at 20 °C for 15 min. The adaptor-ligated DNA was size-selected with NEBNext Sample Purification Beads, and seven-cycle PCR amplification was performed with index primers (i5 and i7) and following PCR conditions: initial denaturation at 98 °C for 30 s, denaturation at 98 °C for 10 s, annealing at 65 °C for 75 s, and final extension at 65 °C for 5 min. On an Agilent 4150 Tape Station (Agilent Technologies, Santa Clara, CA, USA), the PCR-enriched libraries were assessed. The libraries were loaded onto the HiSeq2500 for sequencing and cluster generation to generate 150 bp paired-end reads.

### 2.6. Bioinformatics Analysis

The metagenomic raw reads were assessed for quality and adaptor contamination using FastQC v0.11.9 [22]. The leftover adapters, low-quality bases of Q < 30 and shorter reads (<100 bp), were cleaned using trimmomatic v0.39 [23] using the following parameters: ILLUMINACLIP:TruSeq3-PE-2.fa:2:30:10 SLIDINGWINDOW:15:30 MINLEN:100 TRAILING:30 AVGQUAL:30. The host reads contamination in the data was cleaned by mapping the quality-filtered short reads against the host genome, as suggested by [24]. Accordingly, the clean reads were mapped to reference host genome assemblies ARS-UCD1.2 (RefSeq assembly accession: GCF_002263795.1) for cattle and NDDB_SH_1 (RefSeq assembly accession: GCF_019923935.1) for buffaloes using BowTie2 v2.5.0 [25], and unmapped reads were saved for further analysis. The unmapped reads from the previous step were taxonomically classified based on the K-mer approach using Kraken2 [26], and the full report output was classified into taxonomic levels in Pavian v1.2.0 [27]. The alpha diversity represented by the Shannon index and the beta diversity matrix based on the Bray-Curtis dissimilarity represented using Principal Coordinate Analysis (PCoA) were generated at the genus level in MicrobiomeAnalyst [28]. The data normalisation was performed at different taxonomic ranks using the total sum scaling (TSS) feature in MicrobiomeAnalyst [28], which converted the feature read counts clustered within the same taxonomic rank as a proportion of the total number of reads in the respective sample [29]. The significance of comparative assessment of metagenome composition at taxonomic levels between cattle and buffaloes was performed using the Wilcoxon rank sum test with Benjamini–Hochberg correction (FDR) for the adjusted *p* value of <0.05, OR analysed in MicrobiomeAnalyst using DeSeq2 [30].

### 2.7. Gene Prediction and CAZyme Annotation

The steps in CAZyme prediction involve assembling contigs from the short read sequencing data, predicting open reading frames (ORFs), clustering to reduce redundancy, and mapping against the CAZy database [31] as described previously [32,33,34]. In brief, the paired-end and single-end host contamination-clean reads were assembled in MEGAHIT v1.2.9 [35] with the default parameters, and contigs of <1 kb were removed. Prokaryotic genes were predicted in the contigs using MetaGeneMark [36] using the GeneMark.hmm prokaryotic gene prediction algorithm (version 3.25). The predicted proteins were subjected to CD-Hit v4.8.1 clustering at a sequence identity threshold of 95% to remove redundancy [37]. Candidate CAZymes were identified in the non-redundant protein files using DIAMOND v2.0.15.153 [38], HMMER v3.2.1 [39], and eCAMI (Xu et al., 2020 [40]) tools in a standalone version of the dbCAN annotation tool [41]. The combined output files generated in DIAMOND, HMMER, and eCAMI were filtered, and CAZymes reported in at least two of the above tools were considered for further analysis. The abundance values of the CAZymes were assigned by mapping the clean reads from each sample to the corresponding unigenes obtained after CD-HIT clustering. The mapping process was conducted using the RNASeq analysis tool, as previously described [42], with the exception that it was performed in CLC Genomics Workbench (v21.0.2; QIAGEN, Germantown, MD, USA), and gene abundance in each sample was expressed as transcripts per million (TPM). Subsequently, the abundance values were added to the CAZy predictions to determine the proportion of the different CAZymes present in the rumen metagenome. The CAZymes were categorised into different classes: carbohydrate-binding modules (CBM), carbohydrate esterases (CE), glycoside hydrolases (GH), glycosyl transferases (GT), and polysaccharide lyases (PL).

### 2.8. Protozoal Enumeration

The rumen fluid collected from the cattle and buffaloes for protozoal counts was transported to the laboratory at room temperature and mixed with an equal volume of formal saline. The sample was left at room temperature overnight. The protozoa counting was performed under an Eclipse Ci-S phase-contrast microscope (Nikon Corporation, Tokyo, Japan) as described by [43]. The protozoal counts were expressed as ×10^7^ cells/mL. A method by [44] was used for the morphological identification of the ruminal protozoa.

### 2.9. Enteric CH_4_ Emission

After 30 days of experimental feeding, the sulphur hexafluoride (SF_6_) tracer technique of [45] was employed to quantify the daily enteric CH_4_ emission in cattle and buffaloes. Brass permeation tubes of an appropriate size [10,11,20,46,47,48] were used as sources of SF_6_ in the rumen. The permeation tubes were charged with SF_6_ (731 ± 8.95 mg) in liquid nitrogen. The weekly SF_6_ release rate was recorded over 10 weeks at 39 °C. The mean SF_6_ release rates (mg/d) from the brass permeation tubes were 3.70 ± 0.26. The calibrated brass permeation tubes were inserted into the cattle and buffalo’s rumen 10 days prior to the commencement of the CH_4_ trial. The halters were assembled [49] using nylon tubes, capillary tubes (Supelco, 56712-U, ID 1/16, Sigma Aldrich, St. Louis, MO, USA), and quick connectors (B-QC4-D-200, Swagelok, Solon, OH, USA). The background air sample in the animal shed was collected daily using a PVC canister for gas sampling placed at the ventilated iron wire mesh above the north cemented wall. The canisters connected to the halter assembly were replaced daily at a fixed time to avoid variation in the total collection time. The pressure in the vacuumised and gassed PVC canisters was measured using a digital pressure meter (Leo 2, Keller Druckmesstechnik AG, Winterthur, Switzerland). In the laboratory, the sampled breath and background air in the canisters were diluted (2.50–3.50 folds) with a high-purity N_2_ gas for easy successive sub-sampling. The sub-sampled gas samples were analysed on the gas chromatograph (GC 2010 plus, Shimadzu Corporation, Kyoto, Japan) equipped with a flame ionization detector (FID) and an electron capture detector (ECD) for the estimation of CH_4_ and sulphur hexafluoride gases, respectively. The GC conditions used in the study were those reported previously by [10,20,48]. Six successful minimum collections of breath samples per animal were ensured for a complete feeding cycle of 24 h. For the purpose of calculating daily enteric CH_4_ emission, the equation of [50] was used, whereas CH_4_ yield (g/kg DMI) was calculated by dividing the daily enteric CH_4_ emission (g/d) by the dry matter intake for the individual animals on the particular day.

### 2.10. Chemical Composition, Nutrient Intake, and Digestibility

The chemical composition of the feed ingredients, viz., finger millet straw, para grass, and concentrate, was determined following standard procedures. The moisture content (%) in the feed ingredients was determined as per [51] by drying at 100 °C for 24 h. For ash determination, the samples were ignited in crucibles on a hot plate and thereafter transferred to a muffle furnace for complete oxidation at 550 °C for 4 h [51]. However, the organic matter was calculated by the difference between the initial weight and the ash obtained after oxidation. For determining the crude protein (CP), the nitrogen content in the feed ingredients was ascertained in accordance with [52] by performing digestion, distillation, and titration in an automatic nitrogen analyser (Vepodest 450, Gerhardt, Königswinter, Germany). The CP in feed ingredients was calculated by multiplying the N by 6.25. The crude fibre content (CF) was analysed as per two using an automatic fibre analyser (Fibretherm FT12, Gerhardt, Königswinter, Germany), whereas the neutral detergent fibre (NDF) and acid detergent fibre (ADF) fractions were analysed according to [53].

Concurrent to CH_4_ measurement, a digestibility trial over a seven-day period was conducted in both the cattle and buffaloes. The quantity of dry roughage, green fodder, and concentrate offered to the individual animals was recorded while offering the allowance, whereas the feed residues and faeces voided were recorded the next day (after 24 h). The DM in the offered feed, refusals, and faecal samples were determined as mentioned above under chemical composition. Dry matter intake (DMI) by the individual animals was calculated by the difference between feed allowance and feed refusals and expressed as DMI in kilogram (kg) per day. Similarly, the intake for other nutrients (kg/d) was also calculated, considering the difference between the nutrient quantity in offered feed and refusals. The total weight of dung voided by the individual animals over a period of 24 h was quantitatively collected in a plastic bucket covered with the appropriate lid, and an aliquot equivalent to 1/100th was taken for the DM estimation at 100 °C. Another aliquot of dung, equivalent to 1/1000th of the dung voided, was preserved in 25% sulphuric acid for the ammonia estimation in fresh dung. The dried feed, residues, and dung samples were ground using a cyclotec mill (Foss, Hilleroed, Denmark), and the ground samples were analysed for various chemical constituents as stated previously in this section. For determining the apparent digestibility, the dry matter and other nutrients voided through faeces were also taken into consideration in addition to their concentration in the feed offered and refusals. The apparent digestibility of the various nutrients was calculated using the following equation and expressed as %.
$$Apparent\;digestibility\left(\%\right)=\;\frac{Nutrient\;intake\;-\;Excretion\;of\;nutrient_{\;}}{Intake\;of\;nutrient\;}\times 100$$

### 2.11. VFA and Ammonia

The preserved supernatant samples for VFA and ammonia nitrogen were thawed at room temperature. After a short spin, the thawed samples (0.5 mL) were transferred to the autosampler vials (1.5 mL, Agilent, Santa Clara, CA, USA) and loaded into the autosampler coupled with the gas chromatograph (Agilent 7890B, Santa Clara, CA, USA). The concentration of individual VFA was estimated as per Filípek and Dvořák [54], using the gas chromatograph with the following modified conditions [10]: temperature programme: 59 °C–250 °C (20 °C/min, 10 min), injector temperature: 230 °C, and detector temperature: 280 °C.

Individual volatile fatty acids concentration was determined using the following equation:$$VFA\;con.(mmol)=\frac{Peak\;area\;of\;sample\times Conc.\;of\;standard\times dilution}{Peak\;area\;of\;standard}$$

The ammonia nitrogen in the processed samples after thawing was analysed according to [55]. Briefly, 1 mL of boric acid solution was placed in the inner chamber of the Conway disc, and an equivalent volume of sodium carbonate was placed in the outer chamber. One mL of supernatant was pipetted opposite the sodium carbonate in the outer chamber, and the disc was covered with a lid, rotated gently for the mixing of the outer chamber solutions, and thereafter left undisturbed for 2 h at room temperature. The ammonia concentration was determined by titration of the mixed outer chamber contents against 0.01 N sulfuric acid. Ammonia-N concentration was determined with the following formula:



$$\mathrm{Ammonia}\;-\;N\;(\mathrm{mg}/\mathrm{dL})\;=\;\mathrm{mL}\;\mathrm{of}\;0.001\;N\;H_{2}{\mathrm{SO}}_{4}\;\times \;14$$


### 2.12. Statistical Analysis

All the data were evaluated for normality (Gaussian) using the built-in Shapiro–Wilk method in GraphPad Prism (v.9.0; GraphPad Software, San Diego, CA, USA). All the animal trait data were analysed in GraphPad Prism using an unpaired t test and a significance level cut-off of 0.05. The superscripts in the tabulated data were placed wherever the differences between the two group means for a given parameter were significant.

## 3. Results

### 3.1. Alpha and Beta Diversity

The plateau of the rarefaction curves for the metagenome (Appendix A) demonstrated that the depth of sequencing was sufficient to cover the rumen microbial species diversity. The alpha diversity measured by the Shannon index was not different (*p* = 0.420) between the cattle and buffaloes (Figure 1A). Similarly, the beta diversity assessed by Bray–Curtis was also comparable (*p* = 0.638) between the rumen metagenomes of cattle and buffaloes (Figure 1B).

### 3.2. Rumen Metagenome

Aggregately, a total of 118.15 million reads were generated from the rumen metagenome of cattle and buffaloes (Appendix A). The total raw reads generated from the ruminal fluid samples of cattle and buffaloes were 58.01 and 60.13 million, respectively. After trimmomatic quality filtration, a total of 2.17 million reads, including 1.02 million in cattle and 1.15 million in buffaloes, were removed. Further, 2.17% of reads were removed from the metagenome due to host contamination (Appendix A). Finally, 35.4 GB of data, including 17.4 GB in cattle and 18.0 GB in buffaloes, were processed for the taxonomic abundances of the rumen microbiota.

In the present study, microbiota affiliated with 43 phyla, 86 classes, 200 orders, 458 families, and 1722 genera were identified (Appendix A). The top 20 phyla based on their abundance are depicted in Figure 2A. At the phylum level, Bacteroidetes, Firmicutes, Proteobacteria, and Actinobacteria were most abundant in both host species. Bacteroidetes alone constituted more than 1/3 of the total rumen microbiota in cattle and buffaloes (Appendix A). However, their distribution was comparable (*p* = 0.982) between the two host species fed on a similar diet consisting of equal proportions of dry roughage, green roughage, and concentrate. Similarly, the distribution of Firmicutes, Proteobacteria, and Actinobacteria was not different between the cattle and buffaloes. The Firmicutes to Bacteroidetes ratio (*F*/*B*) in the rumen metagenome of cattle and buffaloes 0.65 and 0.68, respectively (Appendix A). The F/B ratio between two hosts was comparable (*p* = 0.494). There was a negative correlation between the Bacteroidetes and Firmicutes in cattle (r = −0.754), buffaloes (r = −0.861), and irrespective of the species (r = −0.766).

Archaea affiliated to nine phyla, namely Euryarchaeota, Crenarchaeota, Thaumerarchaeota, Nanoarchaeota, Lokiarchaeota, Korarchaeota, Thermoplasmatota, Micrarchaeota, and Nanohaloarchaeota were detected in the present study and aggregately constituted 3.44 and 3.40% of the rumen microbiota in cattle and buffaloes, respectively (Appendix A). Among the phyla, the Euryarchaeota was most prominent and represented ~95% of the archaeal abundances in both cattle and buffaloes. However, like other prominent bacterial phyla, the abundance of the Euryarchaeota was also comparable (*p* = 0.982) between the two host species.

At the order level, *Bacteroidales* and *Eubacteriales* constituted the two largest fractions of the rumen metagenome, representing 29–30 and 15–15.5% of the microbiota, respectively (Figure 2B). There was no difference in the abundance of *Bacteroidales* and *Eubacteriales* between the two host species (Appendix A). The *Fibrobacterales* were the third largest order in the rumen metagenome, and their distribution between cattle and buffaloes was comparable (*p* = 0.352). Among the archaea, the *Methanobacteriales* were found to be the most abundant methanogens in both cattle and buffaloes. With an overall sixth most abundant microbial order, *Methanobacteriales* constituted 2.9–3.0% of the rumen metagenome. Like other prominent bacterial orders, the distribution of *Methanobacteriales* was also similar between two host species. Methanogens affiliated with the *Methanosarcinales*, *Methanococcales*, *Methanomicrobiales*, and *Methanomassiliicoccales* orders were identified, having an individual contribution of 0.06–0.09% in cattle and buffalo rumen metagenomes. However, their distribution, like that of *Methanobacteriales*, was also similar between the hosts (Appendix A). Methanogens from *Thermococcales*, *Methanotrichales*, *Methanocellales*, and *Archaeoglobales* were also identified, but at a very low frequency of ≤0.04%.

*Prevotella* was found to be the largest bacterial genus in the rumen metagenome of both cattle and buffaloes (Figure 2D). However, their abundance was comparable (*p* = 0.914) between the two host species. *Fibrobacter*, with an average abundance of ~5%, constituted the second largest genus of the ruminal microbiota and were distributed at a similar frequency (*p* = 0.352) in cattle and buffaloes. *Clostridium* and *Bacteroides* were the other two genera abundantly distributed (<3%) in the rumen metagenome of cattle and buffaloes (Appendix A). Some of the bacteria, such as *Pseudobutyrivibrio*, *Methylobacterium*, *Methylophaga*, and *Brachybacterium*, constituted a very minor fraction of the ruminal microbiota and were differently distributed between the cattle and buffaloes (Appendix A).

*Methanobrevibacter* was the most prevalent genus of methanogens, accounted approximately three percent of the rumen metagenome. However, their distribution was not different between two host species. *Methanosphaera* and *Methanosarcina* were the other two important genera of methanogens identified in the rumen metagenomes of both cattle and buffalo, at a prevalence of <0.1% (Appendix A). Similarly, *Methanomethylophilus* and *Methanobacterium* prevalence in the rumen metagenome was also detected, but at a lower frequency of <0.05%. *Crenarchaeota*-related methanogens like *Acidianus*, *Sulfolobus*, *Saccharolobus*, and *Sulfurishphaera* were also found in the rumen metagenomes of both cattle and buffaloes. In a parallel way, *Thaumarchaeota* methanogens from the genera *Nitrosopumilus*, *Nitrosocosmicusi*, and *Nitrososphaera* were also identified in the rumen metagenome.

### 3.3. CAZyme Abundance

The rumen metagenome data were assembled into 1.48 million contigs, including 0.71 million in cattle and 0.77 million in buffaloes. The average contig numbers per sample and N50 were 143,151, 773, and 153,852, 749, respectively, in cattle and buffaloes (Appendix A). CAZymes, primarily accountable for driving the microbial enzyme-based fermentation of the carbohydrate, were affiliated with the five classes, namely CBM, CE, GH, GT, and PL (Figure 3A,B). Overall, the GH class was most abundant, constituting ~70% of the total CAZyme abundances (Appendix A); however, there was no difference (*p* = 0.364) between the two hosts in the abundance of GH CAZymes. The CAZymes affiliated to the GT class were second most abundant in the rumen metagenome of both cattle and buffaloes, but the distribution frequency was comparable (*p* = 0.234) between the hosts. Similarly, the abundance of other CAZymes such as CE, CBM, and PL were also comparable between the cattle and buffaloes (Appendix A).

A total of 142 CAZyme families were identified in the rumen metagenome of cattle, whereas 141 CAZyme families were identified in buffaloes (Appendix A). The CBM1 family remain unidentified in the rumen metagenome of cattle. On the contrary, CBM88 and GT112 were not detected in the buffalo metagenome. The GH class, with an overall occurrence of 75 families, had the highest representation in the rumen metagenome, whereas there were 23 and 27 families affiliated to the GT and CBM classes, respectively. In the metagenome, GH43, GH3, GH13, GT2, and CE1 were the most prominent families (Figure 4A), and their abundances were comparable between the cattle and buffaloes (Appendix A). Among the GH class, GH43, GH13, and GH3 constituted >1/3 of the total abundance (Figure 4B).

### 3.4. Protozoal Population

The numbers (×10^7^ cells/mL) of total protozoa were comparable (*p* = 0.191) between the cattle and buffaloes fed on a similar diet comprising equal proportions of dry roughage, green fodder, and concentrate (Table 1). Similarly, the numbers (×10^7^ cells/mL) of ciliates affiliated to Entodinimorphida (*Entodiniomorphs*, *p* = 0.112) and Vestibuliferida (*Holotrichs*, *p* = 0.170) were also not different between the two host species in our study.

### 3.5. Daily Enteric CH_4_ Emissions and CH_4_ Yield

Results from the study indicated significantly higher (*p* < 0.0001) daily enteric CH_4_ emissions in cattle than in buffaloes (280 vs. 136 g/d, Table 1), whereas the difference in CH_4_ corrected based on unit dry matter intake (CH_4_ yield) did not prove significant (*p* = 0.612) between cattle and buffaloes. Similarly, the methane emissions per 100 kg body weight were also similar in statistical comparison (*p* = 0.125).

### 3.6. Nutrient Intake, Digestibility, and Rumen Fermentation

The daily nutrient intake was significantly higher in cattle than the buffaloes (Table 2). However, the correction of dry matter and organic matter intake at uniform body weight basis (kg/100 kg BW) did not reveal any measurable difference between two host species. Similarly, the apparent nutrient digestibility was comparable between cattle and buffaloes (Table 2). The rumen fermentation substantiated by total volatile fatty acid (TVFA) and ammonia-N (mg/L) also did not reveal any significant difference between the two host species (Table 3). Further, analysis of individual fatty acids such as acetate, propionate, butyrate, etc. revealed a numerically higher concentration in buffaloes, but the difference did not prove significant between cattle and buffaloes.

## 4. Discussion

Diverse rumen microbiota, such as bacteria, archaea, protozoa, fungi, and viruses [56], act in a syntrophic manner and perform functions such as feed fermentation, microbial protein synthesis [57], VFA production [58], and elimination of detrimental gaseous by-products. Due to their role in H_2_ scavenging, methanogens are crucial rumen inhabitants [59]. It has been reported that buffaloes have a greater population of cellulolytic bacteria [13], fewer protozoa [60], and greater nutrient digestibility than cattle [61,62]. Recent research indicates that buffalo have a lower potential for CH_4_ production than cattle [63]. The impact of diet, host, and environment on the composition of rumen microbiota has been demonstrated [64,65,66,67]; however, there are only a few reports suggesting the difference in ruminal microbiota composition between cattle and buffaloes fed on a similar diet [9,68,69]. Due to the confounding effects of diet, environment, host species, and developmental stages, it is extremely difficult to confirm that the variation in rumen microbiota is an inherent characteristic or governed by the aforementioned variables.

The primary objective of the present study was to investigate the effect of host species on the rumen microbiota composition while being fed a similar mixed diet in an identical environment and of the same gender and age. The identified rumen microbiota affiliated with 43 phyla, 86 classes, 200 orders, 458 families, and 1722 genera in this study were consistent with the previous reports in cattle [10,32,70] and buffaloes [56,71]. The dominance of Bacteroidetes and Firmicutes phyla in the rumen metagenome of cattle and buffaloes was in good agreement with the recent reports [12,56,63,72]. These two bacterial phyla are primarily involved in the degradation of complex polysaccharides and VFA production. Bacteroidetes, one of the major utilizers of H_2_ [73], hold strong capabilities for protein and polysaccharide degradation [74,75], whereas Firmicutes are efficient in breaking lignocellulosic complexes and producing H_2_ [73]. The F/B ratio and a negative correlation between these two phyla in the rumen metagenome of cattle and buffaloes are consistent with the previous studies [11,12,76,77]. A comparable distribution of Bacteroidetes and Firmicutes and a similar F/B ratio in the present study led to a non-significant difference in the nutrient digestibility and VFA production between the cattle and buffaloes fed on a mixed diet consisting of equal proportions of dry and green roughages and concentrate. These findings are in good agreement with the recent reports [12,71]. However, a few reports revealed that buffaloes have a greater abundance of Firmicutes and Bacteroidetes than cattle [9,14]. This deviation in the abundances of Firmicutes and Bacteroidetes could be due to the difference in diet, which is known to have a measurable impact on rumen microbiota composition [78,79,80]. The compositional shift in the diet from grain to high fibre led to an increase in the F/B ratio [77]; however, the similar diet fed to cattle and buffaloes might have led to a similar distribution of both the Firmicutes and Bacteroidetes. Similarly, the comparable distribution of Proteobacteria between cattle and buffaloes also suggests that the diet composition rather than the host species has a more pronounced effect on the rumen microbiota composition. The upsurge in the abundance of Proteobacteria is linked with an increase in concentrate [70], and since the diet composition for both hosts was similar, there was no difference in the Proteobacteria detected in the present study.

Previously, our group [10] reported that the methanogens affiliated with Euryarchaeota and Crenarchaeota were only present in the rumen of cattle and buffaloes fed on a diet comprising hybrid Napier grass and concentrate. The greater diversity of methanogens than in our previous study can be attributed to diet composition and the omics approach used for uncovering methanogen diversity. In this study, we relied upon the shotgun metagenomic approach, while in our previous study [10], the diversity of methanogens was explored using 16S rRNA sequencing that read only a specific region of DNA. The prominence of Euryarchaeota in the rumen of cattle and buffaloes is in consonance with the previous reports [10,11,48,64,81]. Their comparable distribution between cattle and buffaloes can be attributed to the similar diet offered to both host species. Like the present study, methanogens from the phylum Crenarchaeota were also previously reported in the rumen [10,82,83]. Therefore, the recent approaches certainly led to a better understanding of rumen methanogen diversity; however, their participation in methanogenesis as well as substrate requirements have yet to be investigated.

A reduction in fibre degradation is also held accountable for less ruminal methanogenesis. Since the animals of both species were fed on the same diet and possess an analogue microbial profile, we have not observed any difference in the fibre degradation between the cattle and buffaloes. The correlation between phylogeny and function is a recurrent theme in rumen microbiology. Analysis of the abundance and categories of CAZymes in the rumens of bovines could thus aid in the characterization of fibre degradation potential. The similar CAZyme profiles in cattle and buffaloes also did not reveal any difference, which is why the fibre degradation was comparable and did not lead to a significant difference in the CH_4_ yield. The GHs are a broad class of enzymes that are involved in the metabolism of xylan, chitin, starch, and cellulose [84]. GH was reported as the largest CAZyme class, accountable for carbohydrate degradation by loosening the cellulose surfaces and pushing the cellulose chain into the catalytic core [85]. The enzymes from the GH class hydrolyze the glycosidic bonds between the sugars or non-sugar moiety [86]. Several previous reports also revealed a high GH profile in Holstein cow rumen [32,87,88] and buffaloes [56,71]. GTs catalyze the activation of oligosaccharides or glycosidic linkages to various receptors, including proteins, nucleic acids, lipids, oligosaccharides, and small molecules [89]; they were the second most prevalent CAZy family in the rumen of cattle and buffaloes (14–15%).

The association between the ruminal protozoa and methanogens is a classic example of syntrophy. An indirect inhibition of rumen methanogenesis is likely to occur if the protozoal population is significantly reduced [90,91], which leads to the obstruction of interspecies H_2_ transfer to the methanogens [47,92,93]. The numbers of total protozoa as well as *Entodiniomorphs* and *Holotrichs* were comparable, and that is why no difference in CH_4_ yield between cattle and buffaloes was reported in the present study.

## 5. Conclusions

Bacteroidetes with a comparable distribution in cattle and buffaloes were a highly abundant bacterial phylum and constituted ~1/3 of the total microbiota. Similarly, another abundant phylum, Firmicutes, was also found to be similar between the two host species. The most abundant Bacteroidetes and Firmicutes phyla were found to be negatively correlated. Among the nine phyla identified in the rumen metagenome of cattle and buffaloes, the Euryarchaeota was the most prominent archaeal phylum, comprising ~95% of the total archaea; however, like dominant bacterial phyla, their distribution was also not different between the two host species. *Methanosarcinales*, *Methanococcales*, *Methanomicrobiales*, and *Methanomassiliicoccales* were also identified in the rumen, but the distribution frequency was very low. *Methanobrevibacter* was the most prevalent methanogens, accounting for ~3 percent of the rumen metagenome. CAZymes affiliated with five classes were identified in the metagenome, where glycoside hydrolases with an overall abundance of ~70% constituted the largest fraction. However, there was no difference in the CAZymes profile of cattle and buffaloes. We also did not observe any difference in feed intake, nutrient digestibility, rumen fermentation, or protozoal population. Therefore, this study established that methane yield remains comparable between the cattle and buffaloes if the diet composition, feed ingredients, environmental conditions, and rumen microbiota composition, including bacteria, archaea, and protozoa, do not vary between the hosts. It appears that diet composition and environmental conditions have a more pronounced effect on the rumen microbiota structure and methane emissions than the host factor.

## Figures and Tables

**Figure 1 microorganisms-12-00047-f001:**
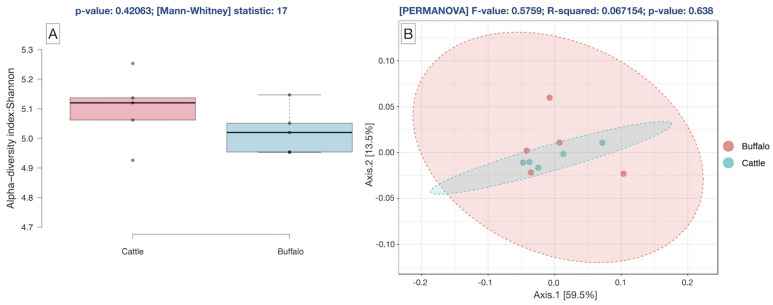
(**A**): Alpha diversity of the cattle and buffaloes rumen metagenome by Shannon index employing Mann–Whitney Kruskal–Wallis Test, (**B**): Beta diversity index of cattle and buffalo rumen metagenome by Bray–Curtis distance method using PERMANOVA.

**Figure 2 microorganisms-12-00047-f002:**
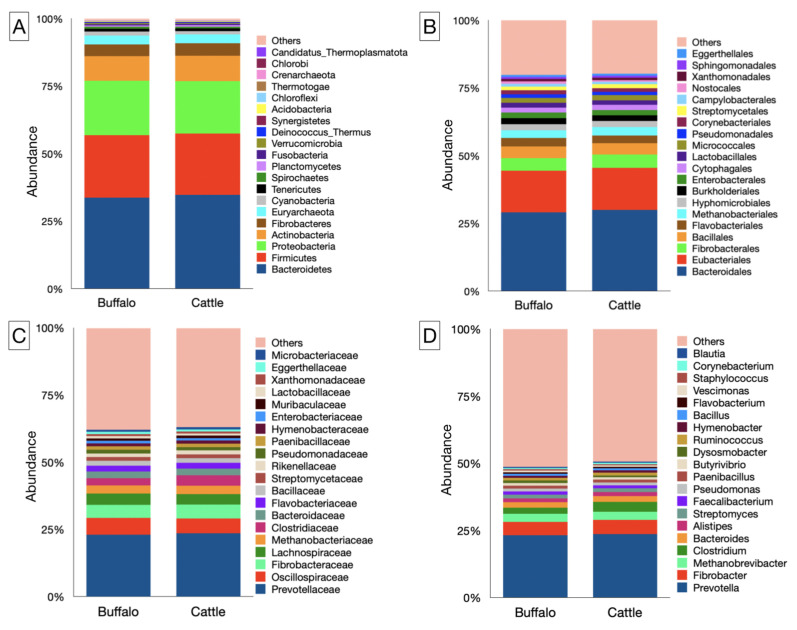
Bacterial and archaeal profiles at different phylogenetic levels. Bar graphs illustrate the relative abundances of rumen microbiota in cattle and buffaloes at the (**A**) phylum, (**B**) order, (**C**) family, and (**D**) genus levels. The bar graphs represent the top 20 microbes at the phylum, order, family, and genus levels.

**Figure 3 microorganisms-12-00047-f003:**
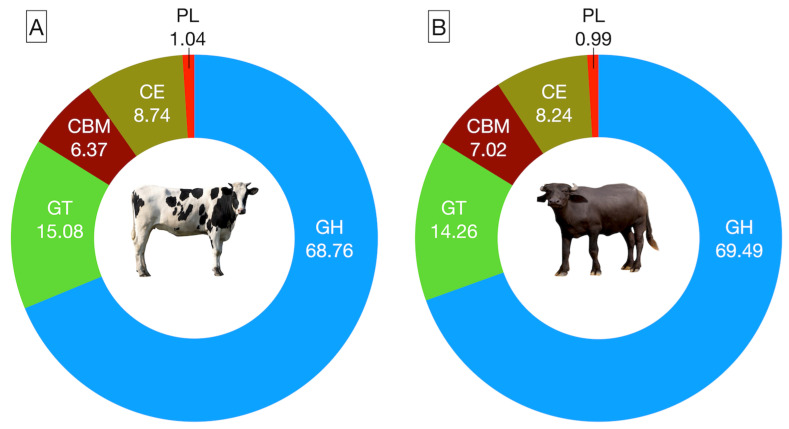
Comparative distribution of CAZymes at the class level in the rumen metagenome of cattle (**A**) and buffaloes (**B**).

**Figure 4 microorganisms-12-00047-f004:**
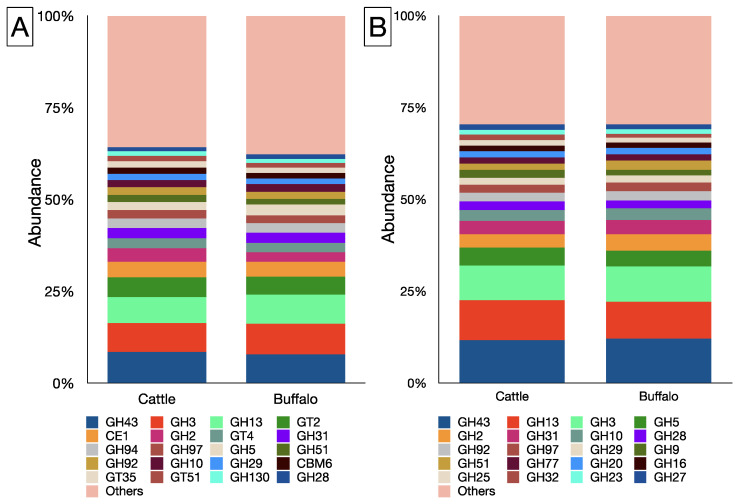
Top 20 CAZyme families of cattle and buffalo metagenome (**A**), top 20 families affiliated to GH class in cattle and buffalo (**B**).

**Table 1 microorganisms-12-00047-t001:** CH_4_ emissions and protozoal population in cattle and buffaloes on a similar diet consisting of equal proportions of dry roughage, green fodder, and concentrate.

Attributes	Cattle	Buffaloes	*p*-Value
Daily enteric CH_4_ (g/d)	280 ± 16.2	136 ± 4.40	<0.0001
CH_4_ (g/100 kg BW)	52.8 ± 3.45	45.0 ± 3.20	0.125
CH_4_ yield (g/kg DMI)	25.1 ± 1.59	23.9 ± 1.10	0.612
Total protozoa (×10^7^ cells/mL)	9.58 ± 0.78	10.8 ± 0.47	0.191
*Entodiniomorphs* (×10^7^ cells/mL)	8.91 ± 0.93	10.7 ± 0.48	0.112
*Holotrichs* (×10^7^ cells/mL)	0.661 ± 0.36	0.131 ± 0.02	0.170

Values in the table are expressed as Mean ± SEM (standard error of mean). Diet comprising finger millet straw, para grass, and concentrate mixture in equal proportions. DMI—dry matter intake, g/d—gram per day.

**Table 2 microorganisms-12-00047-t002:** Nutrient intake, digestibility, and rumen fermentation in cattle and buffaloes when offered a similar diet.

Attribute	Cattle	Buffaloes	*p*-Value
** *Intake* **			
Dry matter (kg/d)	11.2 ± 0.30	5.88 ± 0.53	<0.0001
Dry matter (kg/100 kg BW)	2.11 ± 0.08	1.91 ± 0.13	0.231
Organic matter (kg/d)	10.8 ± 0.28	5.59 ± 0.51	<0.0001
Organic matter (kg/100 kg BW)	2.04 ± 0.10	1.82 ± 0.12	0.191
Crude protein (kg/d)	1.34 ± 0.038	0.620 ± 0.063	<0.0001
Neutral detergent fibre (kg/d)	5.81 ± 0.16	3.18 ± 0.31	<0.0001
Acid detergent fibre (kg/d)	3.23 ± 0.11	1.82 ± 0.19	<0.0001
***Apparent digestibility*** **(%)**			
Dry matter	63.4 ± 1.11	62.3 ± 0.80	0.449
Organic matter	66.3 ± 1.12	65.1 ± 0.54	0.365
Crude protein	70.1 ± 1.16	66.7 ± 1.45	0.095
Neutral detergent fibre	55.0 ± 1.42	54.4 ± 1.09	0.723
Acid detergent fibre	46.5 ± 2.24	49.0 ± 1.65	0.386

Values in the table are expressed as Mean ± SEM (standard error of mean). Diet comprising finger millet straw, para grass, and concentrate mixture in equal proportions. g/d—gram per day, BW—body weight, kg/d—kilogram per day. The mixed diet contained 930 g organic matter, 111 g crude protein, 507 g neutral detergent fibre, 248 g acid detergent fibre, and 69.9 g ash per kg of dry matter. The daily concentrate offered to cattle and buffaloes was 4.45 ± 0.21 and 3.08 ± 0.21 kg, respectively.

**Table 3 microorganisms-12-00047-t003:** Comparative ruminal volatile fatty acid profile in cattle and buffaloes on a uniform diet.

Attributes	Cattle	Buffaloes	*p*-Value
TVFA (mmol)	68.6 ± 4.42	78.3 ± 8.23	0.627
Ammonia (mg/dL)	7.93 ± 1.06	8.87 ± 0.69	0.479
** *Individual VFA* **			
Acetate (mmol)	49.5 ± 2.34	54.3 ± 5.69	0.732
Propionate (mmol)	10.3 ± 2.42	13.2 ± 1.04	0.300
Butyrate (mmol)	4.99 ± 0.88	6.17 ± 0.93	0.554
Iso-butyrate (mmol)	1.44 ± 0.48	1.63 ± 0.51	0.794
Valerate (mmol)	1.42 ± 0.30	1.87 ± 0.17	0.227
Iso-valerate (mmol)	0.881 ± 0.34	1.08 ± 0.36	0.695
A/P ratio	4.70	4.08	0.219

Values in the table are expressed as Mean ± SEM (standard error of mean). TVFA—total volatile fatty acid, A/P—acetate to propionate ratio, mmol—milli mole, mg/L—milligram per litre.

## Data Availability

The datasets presented in this study can be found in online repositories with the accession number PRJNA1015652. The metagenome data with accession number(s) is available in the repository/repositories and can be found at: https://www.ncbi.nlm.nih.gov/sra/PRJNA1015652.

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
