# Peer review of "Comparative Rumen Metagenome and CAZyme Profiles in Cattle and Buffaloes: Implications for Methane Yield and Rumen Fermentation on a Common Diet"

_microorganisms, 2023, doi:10.3390/microorganisms12010047_

Round 1
Reviewer 1 Report
Comments and Suggestions for Authors
Major comments
Authors introduced metagenomic approach to compare rumen microbial
community characteristics of two different species under same ration
component. Although significant difference was not often found, their
dedicated study results would be somewhat worth acknowledged.
Nonetheless their manuscript structure involves substantial pitfalls
such as redundancy which is strongly recommended to put intensive
amendments through the revision process.
Minor comments
L40-68 Almost of all parts in their introduction is not correlated to
primal study object, but a simple summary of preceding studies.
Suggest to make substantial reduction of the part.
L187 Their bioinformatic analysis requires some referencing
information to justify the approach which is much less employed
currently than an orthodox amplicon sequencing approach.
Fig 1B How did they build the ellipsoids covering two respective sort
of plots, particularly in which red one (Buffalo) was irregularly
large compared to blue one.
Figs 2 and 4 The reviewer could not figure out the image as it is
shown because of very poor resolutions. One practical idea is to pick
up less groups up to top 10 (Fig 2) or 20 (Fig 4) without compromising
significance of each data.
Table 1, 2, and 3 To the contrary these tables lack data distribution
in each group or residual error of a measurements either of which
should be obligatory to discuss data reliability
L525-534 and L551-611 As is the case in Introduction, these parts are
neither informative given authors simply made a descriptive review
even they should have focused the phenotypic differences of data or
justification of metagenome approach of which they have applied.
Suggest substantial reduction.
Reviewer 2 Report
Comments and Suggestions for Authors
Dear authors,
After carefully reading your very interesting manuscript, I have but a few very minor suggestions, as follows:
- Line 105: When you state 'crsossbreeds' what do you mean? B. taurus x B. indicus, or different breeds? Please clarify;
- I am missing age of the animals used in the experiment, please include a short sentence on this, for both species;
- Please mention the exact number of kg of concentrated fed to the experimental groups;
- Tables, were all the emissions data corrected per 100 kg BW for the two species? Please clarify this, and if so, mentioned it at the end of the name of each table. If not, please proceed to express your values per 100 kg BW, since the animals from the two species had significantly different live weights (533 kg vs. 308 kg.
More general comments: The manuscript is very well written, with a strong introduction and background, a sound experimental design and statistical approach. Results are well presented and compared with data from the literature.
I consider that the manuscript meets the quality standards to be published, following minor improvements.
